# Establishment and Characterization of a Primary Fibroblast Cell Culture from the Amazonian Manatee (*Trichechus inunguis*)

**DOI:** 10.3390/ani14050686

**Published:** 2024-02-22

**Authors:** Flávia dos Santos Tavares, Cesar Martins, Flávia Karina Delella, Luís Adriano Santos do Nascimento, Angélica Lúcia Figueiredo Rodrigues, Sávia Moreira, Adauto Lima Cardoso, Renata Coelho Rodrigues Noronha

**Affiliations:** 1Laboratório de Genética e Biologia Celular, Centro de Estudos Avançados da Biodiversidade, Instituto de Ciências Biológicas, Universidade Federal do Pará, Belém 66075-110, PA, Brazil; flaviatavares.ft.77@gmail.com (F.d.S.T.); adautolimacardoso@gmail.com (A.L.C.); 2Department of Structural and Functional Biology, Institute of Biosciences at Botucatu, Sao Paulo State University—UNESP, Botucatu 18618-689, SP, Brazil; cesar.martins@unesp.br (C.M.); flavia.delella@unesp.br (F.K.D.); 3Laboratório de Óleos da Amazônia, Universidade Federal do Pará, Belém 66075-110, PA, Brazil; adriansantos@ufpa.br; 4Instituto Biologia e Conservação dos Mamíferos Aquáticos da Amazônia (BioMA), Belém 66077-830, PA, Brazil; angelicabioma@gmail.com (A.L.F.R.); moreira.savia@gmail.com (S.M.)

**Keywords:** Amazon biobank, cryopreservation, conservation, sirenians, aquatic mammals, endangered species, andiroba

## Abstract

**Simple Summary:**

The conservation of endangered wild species such as the Amazon manatee (*Trichechus inunguis*) requires important attention, as the species plays a crucial role in the maintenance and stability of the Amazon ecosystem. Among several areas of scientific research that help in the conservation of species, cryopreservation is a technique that guarantees storing cells at extremely low temperatures for long-term preservation. In this work, we established the first Amazonian manatee cell line (TINsf) from a skin biopsy. We observe its shape and growth rate over time. Furthermore, we also evaluated how Amazonian manatee cells reacted to andiroba seed oil (ASO) and found that low concentrations of ASO do not affect the growth of TINsf cells, but high concentrations cause a reduction in cell proliferation and cell death. Here, we promisingly show that, in addition to establishing Amazonian manatee cell lines, it was possible to guarantee their applicability for future research.

**Abstract:**

The vulnerable status of the Amazon manatee, *Trichechus inunguis*, indicates the need to seek measures to guarantee its conservation. In this context, the cultivation of cells in vitro is a strategy that should at least guarantee the preservation of their genetic material. Thus, we established for the first time a primary culture of Amazonian manatee fibroblasts (TINsf) from a skin biopsy of a young male. Karyotypic analysis of the 3rd, 7th, and 12th passages confirmed the taxonomic identity of the species *T. inunguis* (2n = 56/NF = 92) and indicated that this culture presents genomic stability. Gene and protein expression of vimentin at the 13th passage show the predominant presence of fibroblasts in TINsf. To test the cell line’s responsiveness to materials and demonstrate a possible application of this culture, it was exposed to andiroba seed oil (ASO), and its viability and proliferative capacity were evaluated. ASO demonstrated toxic effects at the highest concentrations and longest exposure times tested, reproducing results observed in human cultures, indicating the applicability of TINsf in toxicological and biotechnological studies. After cryopreservation, the TINsf line maintained its proliferative potential, indicating the establishment of a new culture available for future studies.

## 1. Introduction

The Amazonian manatee (*Trichechus inunguis*, Sirenia: Trichechidae) is an aquatic mammal endemic to the Amazon region, found throughout the entire Amazon basin, with its primary distribution along the Amazon River to its mouth [1]. There are two species of manatees present in Brazil, *T. inunguis* and *T. manatus* (Antillean manatee), both coexisting in sympatry in the estuary of the Amazon basin, with reported cases of hybridization manatees in this region [2,3,4]. Currently, the Amazonian manatee is classified as vulnerable on the Red List of Threatened Species [5], with major threats arising from direct and indirect human activities, such as accidental captures of calves in fishing nets, strandings (especially during the rainy season), water pollution from pesticide and heavy metal use, deforestation of riparian forests due to inappropriate land use for agriculture, dam construction, livestock, and mining [6,7,8].

The establishment of a cryobank for biodiversity preservation is a crucial strategy to conserve the genetic variability of species, especially of threatened species, ensuring their long-term survival. This technique has been widely used in biological and conservation research to store valuable genetic material such as sperm, eggs, and somatic cells, contributing to biodiversity preservation. The process involves the collection, processing, storage, and management of biological material, such as seeds, cells, or tissues, under ultra-freezing conditions. This entire process plays a fundamental role in ex situ conservation and can serve as a safety net against threats such as climate change, diseases, and habitat loss. It is a valuable tool for the long-term preservation of biological diversity [9]. This is, therefore, a good strategy among conservation plans of *T. inunguis*.

For more advanced future research in cell and molecular biology, including the production of biological and genetic therapeutics such as antibodies, interferons, clotting factors, vaccines, toxicological analyses, induction of pluripotent cells, and nuclear transfer in somatic cells (cloning), it is necessary to establish and characterize cell lines [10,11,12,13]. The acquisition of somatic cells as fibroblast lines has been widely established in different representatives of global wildlife, mostly consisting of vertebrates such as mammals, birds, amphibians, and reptiles, especially those facing the threat of extinction [14,15]. Globally, these studies have been increasing in aquatic mammals in recent years [16,17,18,19,20,21], validating the importance of cryopreservation studies, especially for free-living animals susceptible to environmental changes. Mammalian cell culture can be applied in a variety of cellular assays to investigate morphology, protein expression, cell growth, differentiation, apoptosis, and toxicity in different environments, provided that the quality of cell line characterization is ensured [13]. Successful studies on the isolation of cells from *T. manatus* have already been reported. Sweat et al. [22,23] established cell lines from the kidney and bronchi of fresh *T. manatus* carcasses. Twenty years later, Nascimento et al. [24] advanced studies on the use of cryoprotectants for cryopreserving somatic tissues of *T. manatus* to assess their potential effects and establish vitrification protocols for the species.

In order to contribute to the conservation of *T. inunguis,* here, we establish and describe for the first time, a cell culture of this specie. This is a initial step to assist future efforts in the conservation, breeding, and reintroduction of these animals into the wild; to study the biology and physiology of the species in response to environmental stimuli and potential threats to its survival; and to provide opportunities to advance scientific knowledge, veterinary medicine, and public awareness regarding the importance of biodiversity conservation in the Amazon region.

## 2. Materials and Methods

### 2.1. Samples

A 4 mm biopsy was extracted from the caudal fin of a male individual estimated to be six months old (Figure 1a,b) collected at Santa Bárbara do Pará city, State of Pará, Brazil (Figure 1c). The animal is being kept at the Centro Nacional de Pesquisa e Conservação da Biodiversidade Marinha do Norte (CEPNOR), Belém, PA, Brazil. Collections were authorized by the Chico Mendes Institute for Biodiversity Conservation (ICMBIO; Registration: 44915-16) and all the procedures were conducted in accordance with the protocols of the Animal Ethics Committee of the Instituto de Ciências Biológicas of Universidade Federal do Pará (Protocol 8803211223).

### 2.2. Cell Culture Establishment and Cryopreservation

The procedures for the establishment of the primary cell culture were conducted exactly as described in the protocol optimized by Cardoso et al. [25] for ray-finned fishes. The selected caudal fin region was cleaned with 70% alcohol. A 4 mm-diameter biopsy was initially cut using a scalpel; the dermis and epidermis were separated and stored in Dulbecco’s modified Eagle medium: Nutrient F12 (DMEM/F12) supplemented with penicillin (1000 U/mL), streptomycin (1000 μg/mL), and amphotericin B (2.5 μg/mL) at 4 °C until sample processing at the Department of Structural and Functional Biology (Institute of Biosciences at Botucatu, Sao Paulo State University). The dermis biopsy was further cut into small fragments with scissors, and these fragments were washed in sodium hypochlorite (0.2%) for 10 s, ethanol (70%) for 10 s, and sterile 1xPBS (pH = 7.4) for 30 s. Next, the biopsy fragments were treated in DMEM/F12 supplemented with penicillin (1000 U/mL), streptomycin (1000 μg/mL), and amphotericin B (2.5 μg/mL) for 2 h. After, they were transferred to a 15 mL tube containing 1 mL of collagenase type I (1 mg/mL) diluted in DMEM/F12, and the tube was maintained in a water bath at 37 °C for 12 h until dissociation. The dissociated material was centrifuged at 300 g for 10 min, the supernatant was removed, and the pellet was homogenized in 3 mL of DMEM/F12 medium supplemented with 20% fetal bovine serum (FBS), penicillin (100 U/mL), streptomycin (100 μg/mL), and amphotericin B (0.25 μg/mL). The material was transferred to 25 cm^2^ cell culture flasks with a filter and maintained in an humidified incubator at 37 °C and 5% CO_2_ until 90–100% confluence. Cells were trypsinized using the TrypLE^TM^ Express Enzyme 1× for 3 min and transferred to new flasks until reaching confluence. For cryopreservation, cells at 90–100% confluence were trypsinized and centrifuged and the pellet (~2 million cells) was resuspended in 1 mL of DMEM/F12 + GlutaMAX^TM^ medium with 10% FBS and 10% DMSO. The cell suspension was slowly cooled at a rate of 1 °C/min to −20 °C and −80 °C, respectively, for 24 h each, and then stored in liquid nitrogen. For revival, cells were thawed in a water bath at 37 °C, transferred to a 15 mL centrifuge tube containing 3 mL of complete medium, and centrifuged at 300× *g*. The supernatant was removed, the cell pellet was resuspended in 4 mL of complete medium, and cells were transferred to a cell culture flask to grow.

### 2.3. Cell Growth Curve

At the 7th passage, the cell growth curve was generated. Cells were seeded in plates of 24 wells (1.5 × 10^4^ cells per well) and incubated at 37 °C and 5% CO_2_. Every 24 h, the average cell density of three wells was determined using a hemocytometer. The population doubling time (PDT) was calculated with the formula: PDT = T ln2/ln(Xe/Xb), T is the incubation time in hours, Xb is the cell number at the beginning of the incubation time, and Xe is the cell number at the end of the incubation time.

### 2.4. Karyotyping

At the 3rd, 7th, and 12th passages, metaphase chromosomes were obtained. Cells were treated with KaryoMAX^TM^ Colcemid^TM^ (10 ng/mL) in DMEM/F12 medium for 15 min at 37 °C, followed by hypotonization in 0.075 M KCl for 30 min at 37 °C and cell fixation in fresh Carnoy solution. Fixed cell suspensions were dropped onto slides and these were stained with Giemsa. Chromosomes were observed in an optic microscope and photomicrographs were obtained for chromosome counting. The chromosome number of at least thirty cells was determined in each passage analyzed.

### 2.5. Immunocytochemistry

Cells were plated on a coverslip placed at the bottom of a six-well plate and cultured until reaching 80% confluence. The experiment was conducted in triplicate. At this point, the medium was removed and the cells were washed with 1× phosphate-buffered saline (1xPBS) and fixed with formaldehyde (10%) for 8 min at room temperature. Then, they were washed twice with 1xPBS and treated with 1xPBS + 0.1% Triton for 10 min. Then, blocking was carried out with 3% bovine serum albumin (BSA) for 40 min at room temperature, followed by two washes with 1xPBS. The anti-vimentin antibody (Abcam, Cambridge, UK; AB92547) was applied in the ratio of 1 antibody:200 1% BSA and kept for 1 h at room temperature. Two washes with 1xPBS were performed and detection of the primary antibody was performed with Goat Anti-Rabbit IgG H&L (Abcam, Alexa Fluor^®^ 488; AB150077) in the ratio of 1 antibody:200 1% BSA and kept for 1 h at room temperature. Two washes with 1xPBS were performed and the coverslip was transferred to a glass slide, where it was stained with Fluoroshield with DAPI (Sigma-Aldrich, St. Louis, MO, USA; F6057). The cells were observed and photographed under a fluorescence microscope.

### 2.6. Reverse Transcription and Polymerase Chain Reaction (RT-PCR)

At the 13th passage, the total RNA was extracted with the PureLink RNA Mini Kit (Thermo Fisher Scientific, Waltham, MA, USA; 12183018A) and treated with DNase I (Thermo Fisher Scientific, EN0521) following the manufacturers’ instructions. Complementary DNA (cDNA) was synthesized with the High Capacity cDNA Reverse Transcription Kit (Thermo Fisher Scientific, 4368814). Primers for PCR were selected from a previous work based on the human vimentin gene sequence [26]. These primers showed 100% of alignment with the Western Indian Manatee vimentin gene and, therefore, were used here. PCR assays using 100 ng of cDNA as a template were conducted with Taq DNA Polymerase Master Mix RED (Ampliqon, Odense, Denmark, A180301) and 400 nM of each primer in a final volume of 15 μL. The PCR product was observed by 2% agarose gel electrophoresis.

### 2.7. Cell Viability after Andiroba Seed Oil Treatment

Andiroba seed oil (ASO) is recognized in traditional medicine for several properties, including wound healing [27], and therefore it was used here to test the effects of this compound on the regeneration capacity of the Amazonian manatee cell culture. Cells were seeded in a 96-well plate (5 × 10^3^ cells/well) in a volume of 100 µL of DMEM (Dulbecco’s modified Eagle medium), supplemented with 10% FBS, penicillin (100 U/mL), streptomycin (100 μg/mL), and amphotericin B (0.25 μg/mL), and cultivated at 37 °C and 5% CO_2_. The cells were maintained in culture for 24 h. Afterwards, another 100 µL of supplemented culture medium containing the treatments was added. The control treatment was carried out with dimethyl sulfoxide (DMSO) added to the supplemented medium at a rate of 0.87%. ASO was first dissolved in DMSO (0.87%) and then diluted in the supplemented culture medium. The concentrations used were 9.76 μg/mL, 19.53 μg/mL, 39.06 μg /mL, 78.12 μg/mL, 156.25 μg/mL, 312.5 μg/mL, 625 μg/mL, 1250 μg/mL, 2500 μg/mL, and 5000 μg/mL. Treatments were maintained for 24 h with five replicates for each group. Cell viability was assessed by the MTT (tetrazoline 3-(4,5-dimethylthiazol-2yl)-2,5-diphenyl bromide) assay. The treatment medium was removed and 100 µL of medium containing MTT (0.5 mg/mL) was added to each well and maintained for 3 h. The medium with MTT was removed and 100 µL of DMSO was added per well. After 1 h, the absorbance reading was performed with a 570 nm filter on a spectrophotometer. The absorbance intensities were transformed into percentages and the absorbance of the control treatment was defined as 100%. The experiments were conducted in five technical replicates and three biological replicates. For comparison between groups, the ANOVA test was applied followed by the Tukey test, adopting a significance level of 5%.

### 2.8. Scratch Wound Assay

Cells were plated on a six-well plate (5 × 10^4^ cells per well) and cultured in 100 µL of DMEM supplemented with 10% FBS, penicillin (100 U/mL), streptomycin (100 μg/mL), and amphotericin B (0.25 μg/mL) and cultivated at 37 °C and 5% CO_2_ until confluence. Single scratches were made in confluent cell monolayers using 10 μL pipette tips and the medium was replaced. Three groups were evaluated: negative control (NC), ASO at 10 μg/mL, and ASO at 150 μg/mL. The experiment was carried out in triplicate. Every 24 h postwounding, the wounds were photomicrographed in an inverted microscope until the experiment reached 72 h. The perimeter of each wound was traced using ImageJ 1.54d software (https://imagej.nih.gov/ij/ (accessed on 16 October 2023)), and the area of every wound at each time point was then normalized to its respective area at time 0 (Appendix A).

## 3. Results

After biopsy dissociation and plating, cells attached to the surface of culture flask before the 24 h, and confluence was achieved after 10 days. At the 2nd passage, cells were cryopreserved and after two weeks were thawed. At the 7th passage (Figure 2a), the growth curve of TINsf cells was generated, which exhibited a lag stage until 2 days after plating, followed by increases in the growth rate after this time (Figure 2f). The PDT calculated from the curve was 63.78 h.

At the 3rd passage, the cell line demonstrated a karyotype with a diploid number (2n) = 56 chromosomes, (FN) = 92 and karyotypic formula (KF) = 19 m-sm/8a (Figure 2g). At the 7th and 12th passages, TINsf cells showed the same karyotype constitution (Appendix A).

RT-PCR (Figure 2e) and immunohistochemistry (Figure 2b–d) showed the presence of vimentin RNA and protein, respectively, at the 13th passage of TINsf cells.

The MTT assay showed that up to 1250 μg/mL ASO does not promote effects on cell viability; however, from 2500 μg/mL there is an is observable reduction in cell proliferation (Figure 3a). Based on these results, three concentrations (10 μg/mL, 150 μg/mL, and 5000 μg/mL) were selected for the scratch wound assay. In this test, no effect was observed from 10 μg/mL ASO in the wound closure, while it was registered that 150 μg/mL ASO promoted a reduction in cell migration at 72 h (Figure 3b,c). In turn, the 5000 μg/mL ASO led to cell death within a few hours, which caused cell detachment and made it impossible to evaluate this concentration in this test (Appendix A).

## 4. Discussion

In the present work, we report the development of a primary cell culture for the Amazonian manatee *Trichechus inunguis* (TINsf). Similar approaches have been employed in aquatic mammals to establish primary cell lines, particularly for threatened species. Obtaining samples for such studies involves a distinct perspective. Generally, tissue sampling from different locations of the animal, particularly from post-mortem individuals, has proven efficient for these studies [21,22,23,28]. In previous studies on manatees, cell lines were derived from biopsies obtained from the fresh carcasses and internal organs of West Indian manatees (*Trichechus manatus latirostris*) (Table 1), demonstrating similar efficiency. In this study, we collected a biopsy from the caudal fin of a live individual, similar to the non-invasive method employed by Nascimento et al. [24], ensuring donor survival and wound healing within twenty days.

The protocol used for this was exactly the same as that optimized and described by Cardoso et al. [25] to obtain cell cultures from a wide range of ray-finned fishes, expanding the use of this method to other vertebrates. The methodologies used in studies with *T. manatus* are based on the explant method [22,23,24]. Here, the culture was carried out using the enzymatic method. This choice was made due to observations that the cell expansion process occurs more quickly, when compared to the explant method [25]. However, the TDF of TINsf (63.78 h) was higher than that observed in the cell cultures established for *T. manatus* [22,23]. This is surprising, coupled with the fact that the donor individual is quite young. However, this difference may be related to interspecific physiological variations and also to differences in the cultivation methods and organ of origin of the cells used.

The karyotype observed in TINsf is exactly the same described for *T. inunguis* [4,29,30] in all passages assessed, indicating genome stability along the cell growth. Moreover, the karyotype confirms the taxonomic identity of the sample (*T. inunguis*), since it was collected in a region recognized for the occurrence of hybridization between *T. inunguis* and *T. manatus.* Hybrid manatees individuals found in this region exhibited diploid numbers of 49 and 50 chromosomes, respectively [2,4].

The morphology of TINsf cells showed a predominant fusiform shape in the 13th passages studied. These cells provide the structural framework in many tissues and play an essential role in wound healing and tissue repair [31]. During cell growth, observations suggest the presence of several cell types, which is commonly observed in primary cultures [28], although Mehrabani et al. [32] found only fibroblasts from the 2nd cell culture passages onwards. In order to characterize the established culture, markers for vimentin were used, demonstrating that most cells present in this line are TINsf fibroblasts (Figure 2a–e). This is important to assess, as occasional cell differentiation can occur based on growth patterns in different organisms. Despite humpback whale (*Megaptera novaeangliae*) cells [18] maintaining standardized fibroblast morphology up to the 30th passage, confirmed by vimentin expression in cells, primary cells of the pygmy killer whale (*Feresa attenuata*) [21] showed changes in cellular morphology from the 14th passage. Additionally, it is noteworthy that puma (*Puma concolor*) cell lines exhibited ultrastructural changes from the 10th passage, although vimentin presence in cells was confirmed [33].

Cell cultures have diverse applications and are extremely important for the conservation of threatened species [33,34,35], such as manatee species. The establishment and characterization of primary cell lines are an important initial step for research in biotechnological and ecotoxicological applications to address crucial questions about conserving the genetic heritage of wildlife, the susceptibility of these species in the environment, and how they are affected [17]. From a biotechnological perspective, establishing and characterizing manatee cell lines can make them an effective resource for somatic cell nuclear transfer experiments [12,36], enhancing their efficiency in responding to cell viability, both for their own maintenance through passages, growth phases, cryopreservation, and recovery after thawing, as well as for the future use of these induced cell lines in toxicological analyses and gene therapies [13,19,37]. Yajing et al. [21] found that transfected cells from the pygmy killer whale (*Feresa attenuata*) exhibited a higher cell proliferation rate (68.9 h) than primary cells obtained (14.4 h). Furthermore, recent advances in applying assisted reproductive technology (ART) promise viable solutions to prevent the loss of threatened species such as the case of the northern white rhinoceros (*Ceratotherium simum cottoni*) [38,39], and studies presented by Katayama et al. [36] that induced the immortalization of Okinawa rail (*Gallirallus okinawae*) fibroblast cells, an endemic specie of Japan. From an ecotoxicological perspective, it has become important to observe how TINsf can respond to environmental conditions in aquatic environments. This has already been addressed in various aquatic mammals, mainly to understand how the effects of aquatic pollution directly impact these species [17,18,21], such as studies conducted on cells obtained from fin whales (*Balaenoptera physalus*) [40], humpback whales (*Megaptera novaeangliae*) [18], and killer whales (*Orcinus orca*) [16], which have shown cytotoxic and genotoxic effects of environmental pollutants on the cell lines of these species.

Therefore, aiming to evaluate these properties in TINsf, as well as the response of this lineage to compounds, we exposed these cells to ASO, which has previously reported healing properties [27]. The MTT assay showed that high concentrations of ASO decrease cell viability within 24 h of exposure. In turn, the healing test showed a reduction in cell migration after 72 h of exposure to 150 μg/mL, which indicates that the continuous and prolonged use of ASO is not appropriate. Furthermore, no positive effect of andiroba was observed in the concentrations and exposure times analyzed. Similar results were observed after exposure of the ACP02 strain to ASO [41], which indicates that the TINsf cell line reflects the biological responses observed in cells from other mammals, and, therefore, TINsf can be used in toxicological and biotechnological studies.

## 5. Conclusions

For the first time, a cell culture was developed and characterized for the threatened Amazonian manatee. The methodology used has a high value for use in conservation plans, as it allows the derivation of cultures in a non-invasive way and without compromising the lives of individuals. The culture-based karyotyping is a highly efficient method for confirming the taxonomic identity of individuals, especially in hybridization zones. The cell culture obtained has the potential for use in toxicological studies and biotechnological applications and represents a relevant source for obtaining chromosomes, DNA, RNA, proteins, and other biomolecules. Furthermore, we demonstrated that an optimized protocol for cell culture in ray-finned fish was highly efficient for use in a mammalian species, and that it could become standard for vertebrates.

## Figures and Tables

**Figure 1 animals-14-00686-f001:**
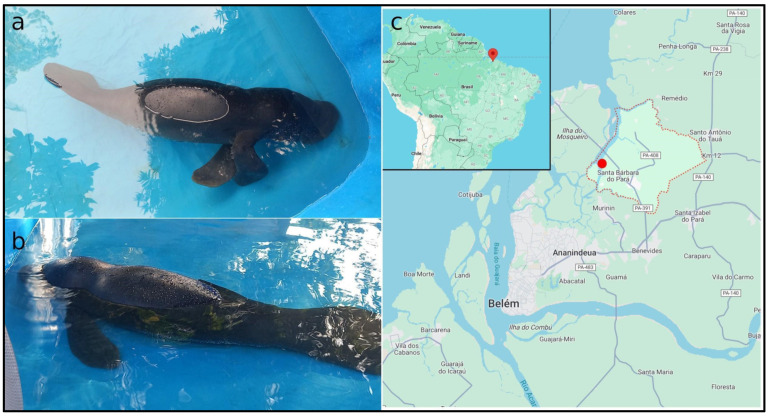
The Amazonian manatee *Trichechus inunguis*. (**a**,**b**) Photos of the sample analyzed in the present study; (**c**) Map indicating the site of collection at Santa Bárbara do Pará city, State of Pará, Brazil.

**Figure 2 animals-14-00686-f002:**
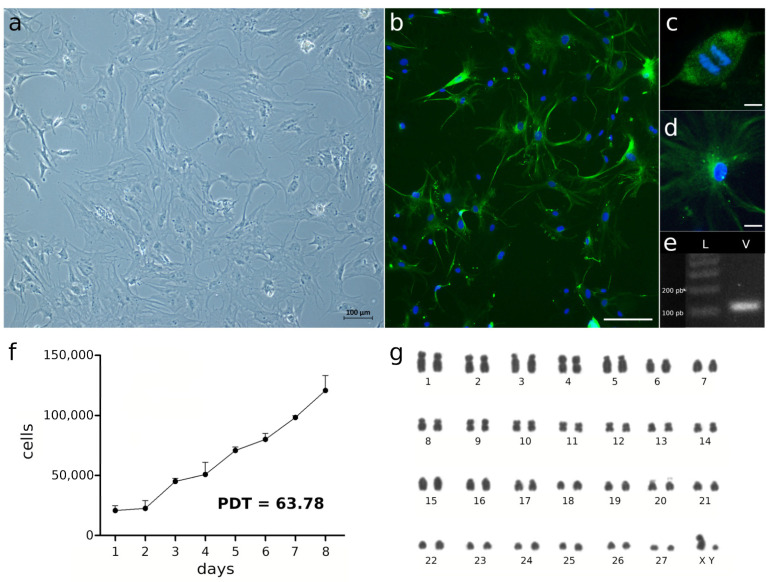
Characterization of the primary fibroblast cell culture from the Amazonian manatee *Trichechus inunguis* (TINsf). (**a**) The cell culture at the 3rd passage; scale bar = 100 μm. (**b**) Expression of vimentin as shown by immunofluorescence with anti-vimentin antibody in TINsf at the 13th passage; vimentin is shown in green and the nuclei are stained with DAPI in blue; scale bar = 100 μm (**c**) A dividing cell highlighted; scale bar = 10 μm. (**d**) A interphase cell highlighted; scale bar = 20 μm. (**e**) Transcription of vimentin gene as assessed by RT-PCR; L—DNA ladder, V—vimentin. (**f**) Cell growth curve; PDT—Population Doubling Time. (**g**) karyotype at the 3rd passage showing 2n = 56.

**Figure 3 animals-14-00686-f003:**
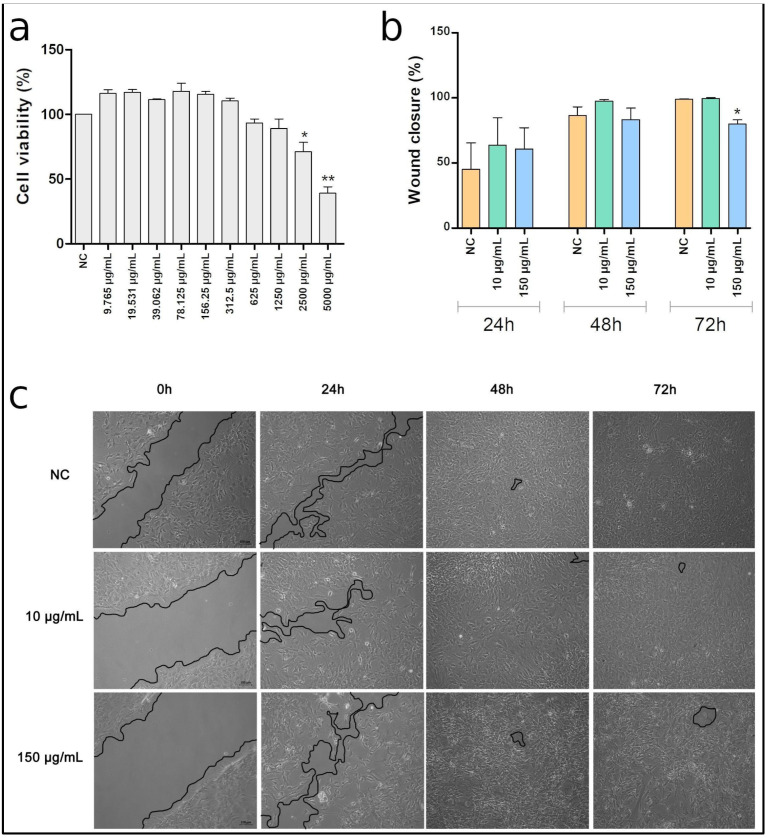
Effects of Andiroba seed oil (ASO) in the TINsf cell line. (**a**) Cell viability as measured by MTT assay showing the toxic effects of ASO from 1500 μg/mL. (**b**) Summary bar graph illustrating the percentage wound closure at the indicated time points during the scratch wound assay. (**c**) Representative images from in vitro scratch wound healing assays demonstrating that cell migration into the cell-free region (outlined) is significantly reduced in the presence of 150 μg/mL ASO when compared to the control. * *p* < 0.05; ** *p* < 0.005.

**Table 1 animals-14-00686-t001:** Cell culture in Sirenia. Compilation of studies that led to the establishment of primary cell cultures.

Species	Cell Name/Source Organ	Reference
*Trichechus inunguis*	TINsf/Skin	Present work
*Trichechus manatus latirostris*	MK/Kidney	Sweat et al. [22]
*Trichechus manatus latirostris*	MRTEC/Bronchi	Sweat et al. [23]
*Trichechus manatus manatus*	-/Dermis	Nascimento et al. [24]

## Data Availability

All datasets which support the findings of this study are available in the manuscript and Appendix A. Additional information or materials may be requested from the corresponding author.

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
