# Peer review of "Establishment and Characterization of a Primary Fibroblast Cell Culture from the Amazonian Manatee (Trichechus inunguis)"

_animals, 2024, doi:10.3390/ani14050686_

Round 1
Reviewer 1 Report
Comments and Suggestions for Authors
Tavares et al. report the creation of a somatic cell line from a skin biopsy of the Amazon manatee. They further provide some cell line characterization and toxicology tests.
Characterization and toxicology are rather basic, but the results are mostly well presented and conclusions drawn are supported by the results.
The only result that is not convincing to me is the scratch wound assay. In the example images shown in Fig 3C, it is not clear, why the authors outline a cell-free spot in the condition 150 ug/mL at 72h in a position where the scratch wound seemingly had already been closed after 48h. Based on the images provided it would have been possible to draw such an outline e.g. also at NC/72h. From the images provided it seems that wounds in all 3 conditions are practically closed after 48h. In this context, it is also not clear to me, what Figure S1 actually depicts.
In addition to this, I have two technical points:
The cryopreservation procedure is not well described. The number of cells and the volume that is cryopreserved are not mentioned. Neither is the actual procedure of cryopreservation described, i.e. how are the cells cooled to liquid nitrogen temperature.
In the methods section it is stated that the growth curve was acquired at the 7th passage, while the results section mentions the 3rd passage.
Comments on the Quality of English LanguageThe english is well understandable but contains some grammatical errors.
Author Response
Tavares et al. report the creation of a somatic cell line from a skin biopsy of the Amazon manatee. They further provide some cell line characterization and toxicology tests.
Characterization and toxicology are rather basic, but the results are mostly well presented and conclusions drawn are supported by the results.
The only result that is not convincing to me is the scratch wound assay. In the example images shown in Fig 3C, it is not clear, why the authors outline a cell-free spot in the condition 150 ug/mL at 72h in a position where the scratch wound seemingly had already been closed after 48h. Based on the images provided it would have been possible to draw such an outline e.g. also at NC/72h. From the images provided it seems that wounds in all 3 conditions are practically closed after 48h. In this context, it is also not clear to me, what Figure S1 actually depicts.
Answer to comment: Firstly, we appreciate your review of this work. Based on the results we obtained, we conclude that the concentration of 150 µg/mL of andiroba seed oil (ASO) caused cell death within a 72-hour exposure time, supporting the reopening of the wound within 72 hours. Additionally, we also presume that ASO concentrations, although low, can induce cell death and a subsequent reduction in cell migration at longer exposure times.
In addition to this, I have two technical points:
The cryopreservation procedure is not well described. The number of cells and the volume that is cryopreserved are not mentioned. Neither is the actual procedure of cryopreservation described, i.e. how are the cells cooled to liquid nitrogen temperature.
Answer to comment: Thank you for pointing out this important aspect. We have made the necessary adjustments to the manuscript.
In the methods section it is stated that the growth curve was acquired at the 7th passage, while the results section mentions the 3rd passage.
Answer to comment: Thank you for pointing out this important aspect. We have made the necessary adjustments to the manuscript.
REVIEWER 1
Comments and Suggestions for Authors
Tavares et al. report the creation of a somatic cell line from a skin biopsy of the Amazon manatee. They further provide some cell line characterization and toxicology tests.
Characterization and toxicology are rather basic, but the results are mostly well presented and conclusions drawn are supported by the results.
The only result that is not convincing to me is the scratch wound assay. In the example images shown in Fig 3C, it is not clear, why the authors outline a cell-free spot in the condition 150 ug/mL at 72h in a position where the scratch wound seemingly had already been closed after 48h. Based on the images provided it would have been possible to draw such an outline e.g. also at NC/72h. From the images provided it seems that wounds in all 3 conditions are practically closed after 48h. In this context, it is also not clear to me, what Figure S1 actually depicts.
Answer to comment: Firstly, we appreciate your review of this work. Based on the results we obtained, we conclude that the concentration of 150 µg/mL of andiroba seed oil (ASO) caused cell death within a 72-hour exposure time, supporting the reopening of the wound within 72 hours. Additionally, we also presume that ASO concentrations, although low, can induce cell death and a subsequent reduction in cell migration at longer exposure times.
In addition to this, I have two technical points:
The cryopreservation procedure is not well described. The number of cells and the volume that is cryopreserved are not mentioned. Neither is the actual procedure of cryopreservation described, i.e. how are the cells cooled to liquid nitrogen temperature.
Answer to comment: Thank you for pointing out this important aspect. We have made the necessary adjustments to the manuscript.
In the methods section it is stated that the growth curve was acquired at the 7th passage, while the results section mentions the 3rd passage.
Answer to comment: Thank you for pointing out this important aspect. We have made the necessary adjustments to the manuscript.

Reviewer 2 Report
Comments and Suggestions for Authors
The manuscript “Establishment of a Primary Continuous Fibroblast Cell Culture from the Amazonian Manatee (Trichechus inunguis)” aimed to obtain fibroblasts from a species of Amazonian manatee, evaluating cellular conditions throughout the passages. The data is interesting, but some points deserve attention.
General comments:
1. Throughout the manuscript, the authors use a very exact approach to interpreting the results, when in fact having only a n sample of 1 animal, some information could be softened.
2. Regarding the experimental n, this parameter is a point that concerns me in the manuscript, as the authors carry out the entire experiment using samples from one animal. How would you justify this?
3. It is not clear to this reviewer the cell toxicity assay to ASO and what this can contribute to the characterization of fibroblasts. In my view, the toxicity testing of these cells is not clear information at this time.
4. In general, the results are not clear, i.e., was only one biological replicate and one technical replicate performed? What are the reasons for the experimental design choices?
Title, and simple summary:
1. What do the authors mean by continuous primary culture? This term seems to remind me of obtaining an immortalized lineage, which would not be the case in this study. I suggest the authors remove this term "continuous" from the title and manuscript.
2. Delete “small” in small skin biopsy.
Introduction:
1. How do the Amazon manatee play a crucial role in the ecosystem it inhabits? Explain this with examples.
2. The introduction needs to discuss the study that resulted in the elaboration of the experimental design, that is, a) How were the passages to be evaluated established? b) Why would we expect a change or not in genetic stability? c) Why would you want to confirm that the cells to be recovered are fibroblasts? Therefore, the authors need to present 2 to 3 paragraphs presenting the study's problematization in a more objective way. In general, the introduction is superficial and unscientific.
Material and methods:
1. Explain further how the skin was harvested: sample size, pre, during and post-harvest conditions performed on the animal, how the tissue was removed, processing conditions in the field and in the laboratory, transport conditions (time, temperature and medium ), between others.
2. Correct "1,500 rpm" for value and unit in g.
3. Why has each analysis generally occurred in different passages? Explain how the passages for the analyzes were chosen.
4. How were toxicity assay concentrations established? It is necessary to better explain the reason for this test.
5. From the skin biopsy, how many primary culture plates were prepared? How many fragments were generated, how many technical replications were carried out?
Results, Discussion and conclusions:
1. As the authors state about genomic stability throughout the passages, I suggest cell karyotypes for each passage are analyzed.
2. Fig 3a is not clear: how can viability result in more than 100%?
3. In these cases, biopsies were obtained from fresh carcasses and internal organs were used, which indicates that the use of these organs are unfeasible for conservation strategies, the authors need to review this statement, as Nascimento et al also harvested the skin (dermis) non-invasively, as in the present study. Also, how did the authors determine “tail” as the biopsy region?
4. In “In this sense, here we collected a tail biopsy from a living individual, which is a non-invasive method that guarantees the survival of the donor and wound healing in a few weeks”, The authors did not present these results, to know whether this statement is correct or not.
5. In my view, the results and discussion need to be re-written. The authors need to improve and present all results, detail the technical replicates (which analyzes were carried out and how many), discuss the results with different groups of mammalian mammals, justify why they used enzymatic disaggregation, etc.
6. In “especially in hybridization zones.”, but the authors do not discuss this throughout the manuscript.
Author Response
The manuscript “Establishment of a Primary Continuous Fibroblast Cell Culture from the Amazonian Manatee (Trichechus inunguis)” aimed to obtain fibroblasts from a species of Amazonian manatee, evaluating cellular conditions throughout the passages. The data is interesting, but some points deserve attention.
General comments:
- Throughout the manuscript, the authors use a very exact approach to interpreting the results, when in fact having only a n sample of 1 animal, some information could be softened.
Answer: Thank you for the observation. Adjustments have been made to the manuscript.
- Regarding the experimental n, this parameter is a point that concerns me in the manuscript, as the authors carry out the entire experiment using samples from one animal. How would you justify this?
Answer: Thank you for noting this point. We understand the concern regarding the experimental number. Given that it involves the Amazonian manatee (an endangered species), we face significant limitations in conducting this type of study. The first obstacle is sample acquisition; for such studies, we rely on partnerships with wildlife rehabilitation centers that assist in rescuing stranded manatees in the estuary of the Amazon, which can occur sporadically. The second point is obtaining a suitable biopsy to make the cells viable for primary culture. We have implemented various protocols in the past but had not succeeded in establishing a viable cell line until we tested the protocol by Cardoso et al. (2023).
REVIEWER 2
Comments and Suggestions for Authors
The manuscript “Establishment of a Primary Continuous Fibroblast Cell Culture from the Amazonian Manatee (Trichechus inunguis)” aimed to obtain fibroblasts from a species of Amazonian manatee, evaluating cellular conditions throughout the passages. The data is interesting, but some points deserve attention.
General comments:
- Throughout the manuscript, the authors use a very exact approach to interpreting the results, when in fact having only a n sample of 1 animal, some information could be softened.
Answer: Thank you for the observation. Adjustments have been made to the manuscript.
- Regarding the experimental n, this parameter is a point that concerns me in the manuscript, as the authors carry out the entire experiment using samples from one animal. How would you justify this?
Answer: Thank you for noting this point. We understand the concern regarding the experimental number. Given that it involves the Amazonian manatee (an endangered species), we face significant limitations in conducting this type of study. The first obstacle is sample acquisition; for such studies, we rely on partnerships with wildlife rehabilitation centers that assist in rescuing stranded manatees in the estuary of the Amazon, which can occur sporadically. The second point is obtaining a suitable biopsy to make the cells viable for primary culture. We have implemented various protocols in the past but had not succeeded in establishing a viable cell line until we tested the protocol by Cardoso et al. (2023).
- It is not clear to this reviewer the cell toxicity assay to ASO and what this can contribute to the characterization of fibroblasts. In my view, the toxicity testing of these cells is not clear information at this time.
Answer: Thank you for your perspective. We believe that the toxicity assay on the TINsf cell line represents an initial future approach to the applicability of using toxicity tests in primary cells, demonstrating their viability in this type of study compared to the use of commonly employed cell lines.
- In general, the results are not clear, i.e., was only one biological replicate and one technical replicate performed? What are the reasons for the experimental design choices?
Answer: Thank you for the observation. All experiments were conducted with experimental replicates. We have adjusted the text indicating the number of replicates in each experiment.
Title, and simple summary:
- What do the authors mean by continuous primary culture? This term seems to remind me of obtaining an immortalized lineage, which would not be the case in this study. I suggest the authors remove this term "continuous" from the title and manuscript.
Answer: Thank you for the observation. We have made adjustments to the manuscript.
- Delete “small” in small skin biopsy.
Answer: Thank you for the suggestion. We have made adjustments to the manuscript.
Introduction:
- How do the Amazon manatee play a crucial role in the ecosystem it inhabits? Explain this with examples.
Answer: Thank you for the observation. We have adjusted the statement in the manuscript for better clarity.
- The introduction needs to discuss the study that resulted in the elaboration of the experimental design, that is, a) How were the passages to be evaluated established? b) Why would we expect a change or not in genetic stability? c) Why would you want to confirm that the cells to be recovered are fibroblasts? Therefore, the authors need to present 2 to 3 paragraphs presenting the study's problematization in a more objective way. In general, the introduction is superficial and unscientific.
Answer: Thank you for the considerations. We have adjusted the Introduction of the manuscript.
Material and methods:
- Explain further how the skin was harvested: sample size, pre, during and post-harvest conditions performed on the animal, how the tissue was removed, processing conditions in the field and in the laboratory, transport conditions (time, temperature and medium ), between others.
Answer: Thank you for the feedback. We have made adjustments to the manuscript.
- Correct "1,500 rpm" for value and unit in g.
Answer: Thank you for the considerations. We have made adjustments to the manuscript.
- Why has each analysis generally occurred in different passages? Explain how the passages for the analyzes were chosen.
Answer: Thank you for raising this question. We chose these passages (3rd, 7th, and 12th passages) analyzed in the present study as we observed a consistent pattern in cell morphology and growth during cultivation, and no alterations were identified.
- How were toxicity assay concentrations established? It is necessary to better explain the reason for this test.
Answer: Thank you for your considerations. We established the concentrations of ASO according to Porfírio-Dias et al. (2020) with some adaptations. We conducted ASO toxicity tests with serial concentrations, each concentration in quintuplicate. For better clarity of the test, we made adjustments in the manuscript.
- From the skin biopsy, how many primary culture plates were prepared? How many fragments were generated, how many technical replications were carried out?
Answer: We performed the cell culture according to the method described by Cardoso et al. (2023). Thus, the experiment was conducted by dissociating the tissue using Collagenase I (1 mg/mL). We started with one culture replica and successfully observed adherent cells within the first 24 hours.
Results, Discussion and conclusions:
- As the authors state about genomic stability throughout the passages, I suggest cell karyotypes for each passage are analyzed.
Answer: Thank you for the comment. We selected passages at the beginning (3rd), middle (7th), and end (12th) of the cultivation period, as we believe these points are sufficient to provide the culture's characteristics. If the culture exhibits the same chromosomal features in all these assessed passages, it strongly suggests that the untested intermediate passages also have the same karyotype.
- Fig 3a is not clear: how can viability result in more than 100%?
Answer: Thank you for the question. Cell viability above 100% (value assumed for the control group) is associated with increased cell proliferation caused by some treatment. However, although bars indicating viabilities above 100% for some of the tested concentrations can be observed in the graph (Figure 3a), we did not find statistical support indicating a difference compared to the control group.
- In these cases, biopsies were obtained from fresh carcasses and internal organs were used, which indicates that the use of these organs are unfeasible for conservation strategies, the authors need to review this statement, as Nascimento et al also harvested the skin (dermis) non-invasively, as in the present study. Also, how did the authors determine “tail” as the biopsy region?
Answer: Thank you for the observation. We have made adjustments to the manuscript.
- In “In this sense, here we collected a tail biopsy from a living individual, which is a non-invasive method that guarantees the survival of the donor and wound healing in a few weeks”, The authors did not present these results, to know whether this statement is correct or not.
Answer: Thank you for the observation. We have made adjustments to the manuscript.
- In my view, the results and discussion need to be re-written. The authors need to improve and present all results, detail the technical replicates (which analyzes were carried out and how many), discuss the results with different groups of mammalian mammals, justify why they used enzymatic disaggregation, etc.
Answer: Thank you for the observation. We have made adjustments to the manuscript.
- In “especially in hybridization zones.”, but the authors do not discuss this throughout the manuscript.
Answer: Thank you for the observation. We have made adjustments to the manuscript.
- It is not clear to this reviewer the cell toxicity assay to ASO and what this can contribute to the characterization of fibroblasts. In my view, the toxicity testing of these cells is not clear information at this time.
Answer: Thank you for your perspective. We believe that the toxicity assay on the TINsf cell line represents an initial future approach to the applicability of using toxicity tests in primary cells, demonstrating their viability in this type of study compared to the use of commonly employed cell lines.
- In general, the results are not clear, i.e., was only one biological replicate and one technical replicate performed? What are the reasons for the experimental design choices?
Answer: Thank you for the observation. All experiments were conducted with experimental replicates. We have adjusted the text indicating the number of replicates in each experiment.
Title, and simple summary:
- What do the authors mean by continuous primary culture? This term seems to remind me of obtaining an immortalized lineage, which would not be the case in this study. I suggest the authors remove this term "continuous" from the title and manuscript.
Answer: Thank you for the observation. We have made adjustments to the manuscript.
- Delete “small” in small skin biopsy.
Answer: Thank you for the suggestion. We have made adjustments to the manuscript.
Introduction:
- How do the Amazon manatee play a crucial role in the ecosystem it inhabits? Explain this with examples.
Answer: Thank you for the observation. We have adjusted the statement in the manuscript for better clarity.
- The introduction needs to discuss the study that resulted in the elaboration of the experimental design, that is, a) How were the passages to be evaluated established? b) Why would we expect a change or not in genetic stability? c) Why would you want to confirm that the cells to be recovered are fibroblasts? Therefore, the authors need to present 2 to 3 paragraphs presenting the study's problematization in a more objective way. In general, the introduction is superficial and unscientific.
Answer: Thank you for the considerations. We have adjusted the Introduction of the manuscript.
Material and methods:
- Explain further how the skin was harvested: sample size, pre, during and post-harvest conditions performed on the animal, how the tissue was removed, processing conditions in the field and in the laboratory, transport conditions (time, temperature and medium ), between others.
Answer: Thank you for the feedback. We have made adjustments to the manuscript.
- Correct "1,500 rpm" for value and unit in g.
Answer: Thank you for the considerations. We have made adjustments to the manuscript.
- Why has each analysis generally occurred in different passages? Explain how the passages for the analyzes were chosen.
Answer: Thank you for raising this question. We chose these passages (3rd, 7th, and 12th passages) analyzed in the present study as we observed a consistent pattern in cell morphology and growth during cultivation, and no alterations were identified.
- How were toxicity assay concentrations established? It is necessary to better explain the reason for this test.
Answer: Thank you for your considerations. We established the concentrations of ASO according to Porfírio-Dias et al. (2020) with some adaptations. We conducted ASO toxicity tests with serial concentrations, each concentration in quintuplicate. For better clarity of the test, we made adjustments in the manuscript.
- From the skin biopsy, how many primary culture plates were prepared? How many fragments were generated, how many technical replications were carried out?
Answer: We performed the cell culture according to the method described by Cardoso et al. (2023). Thus, the experiment was conducted by dissociating the tissue using Collagenase I (1 mg/mL). We started with one culture replica and successfully observed adherent cells within the first 24 hours.
Results, Discussion and conclusions:
- As the authors state about genomic stability throughout the passages, I suggest cell karyotypes for each passage are analyzed.
Answer: Thank you for the comment. We selected passages at the beginning (3rd), middle (7th), and end (12th) of the cultivation period, as we believe these points are sufficient to provide the culture's characteristics. If the culture exhibits the same chromosomal features in all these assessed passages, it strongly suggests that the untested intermediate passages also have the same karyotype.
- Fig 3a is not clear: how can viability result in more than 100%?
Answer: Thank you for the question. Cell viability above 100% (value assumed for the control group) is associated with increased cell proliferation caused by some treatment. However, although bars indicating viabilities above 100% for some of the tested concentrations can be observed in the graph (Figure 3a), we did not find statistical support indicating a difference compared to the control group.
- In these cases, biopsies were obtained from fresh carcasses and internal organs were used, which indicates that the use of these organs are unfeasible for conservation strategies, the authors need to review this statement, as Nascimento et al also harvested the skin (dermis) non-invasively, as in the present study. Also, how did the authors determine “tail” as the biopsy region?
Answer: Thank you for the observation. We have made adjustments to the manuscript.
- In “In this sense, here we collected a tail biopsy from a living individual, which is a non-invasive method that guarantees the survival of the donor and wound healing in a few weeks”, The authors did not present these results, to know whether this statement is correct or not.
Answer: Thank you for the observation. We have made adjustments to the manuscript.
- In my view, the results and discussion need to be re-written. The authors need to improve and present all results, detail the technical replicates (which analyzes were carried out and how many), discuss the results with different groups of mammalian mammals, justify why they used enzymatic disaggregation, etc.
Answer: Thank you for the observation. We have made adjustments to the manuscript.
- In “especially in hybridization zones.”, but the authors do not discuss this throughout the manuscript.
Answer: Thank you for the observation. We have made adjustments to the manuscript.
